# Effectiveness of three delivery models for promoting access to pre-exposure prophylaxis in HIV-1 serodiscordant couples in Nigeria

**Morenike Oluwatoyin Folayan**[1], **Sani Aliyu**[2], **Ayodeji Oginni**[3], **Oliver Ezechi**[4], **Grace Kolawole**[5], **Nkiru Ezeama**[6], **Nancin Dadem**[5], **James Anenih**[7], **Matthias Alagi**[7], **Etim Ekanem**[8], **Evaristus Afiadigwe**[6], **Rose Aguolu**[7], **Tinuade Oyebode**[9], **Alero Babalola-Jacobs**[9], **Atiene Sagay**[9], **Chidi Nweneka**[7], **Hadiza Kamofu**[10], **John Idoko**[9]*

1 Department of Child Dental Health, Obafemi Awolowo University, Ile-Ife, Nigeria, 2 Dept. of Infectious Diseases, Cambridge University Hospitals, Cambridge, United Kingdom, 3 Innovative Aid, Abuja, Nigeria, 4 Nigerian Institute of Medical Research, Lagos, Nigeria, 5 Jos University Teaching Hospital, AIDS Prevention Initiative in Nigeria, Jos, Nigeria, 6 Faculty of Medicine, College of Health Sciences, Nnamdi Azikiwe University, Awka, Nnewi, Nigeria, 7 National Agency for the Control of AIDS, Abuja, Nigeria, 8 Department of Obstetrics and Gynaecology, University of Calabar, Calabar, Nigeria, 9 Jos University Teaching Hospital, Jos, Nigeria, 10 Family Health International, Abuja, Nigeria

* jonidoko@yahoo.com

## Abstract

### Objectives

To evaluate the effectiveness of three models for pre-exposure prophylaxis (PrEP) service delivery to HIV-1 serodiscordant couples in Nigeria.

### Methods

297 heterosexual HIV-1 serodiscordant couples were recruited into three PrEP delivery models and followed up for 18 months. The models were i) Outpatient clinic model providing PreP in routine outpatient care; ii) Antiretroviral therapy (ART) clinic model providing PrEP in ART clinics; and iii) Decentralized care model providing PrEP through primary and secondary care centres linked to a tertiary care centre. The primary effectiveness endpoint was incident HIV-1 infection. The HIV incidence before and after the study was compared and the incidence rate ratio computed for each model. Survival analysis was conducted, Cox regression analysis was used to compare the factors that influenced couple retention in each of the models. Kaplan-Meier survival analysis was used to estimate the median retention time (in months) of the study participants in each of the study models, and log-rank test for equality of survival functions was conducted to test for significant differences among the three models.

### Results

There was no significant difference (p>0.05) in the couple retention rates among the three models. At months 3, 6 and 9, adherence of the HIV-1-infected partners to ART was highest

**Funding:** Author who received award JI Grant #
OPP1104917 Name of Funder: Bill & Melinda
Gates Foundation URL of Funder: gates foundation.
org NO - The funders had no role in study design,
data collection and analysis, decision to publish, or
preparation of the manuscript.

**Competing interests:** The authors have declared
that no competing interests exist.

in the decentralized model, whereas at months 9 and 12, the outpatient model had the highest proportion of HIV-1- uninfected partners adhering to PrEP (p<0.001). The HIV incidence per 100 person-years was zero in the general outpatient clinic and ART clinic models and 1.6 (95% CI: 0.04–9.1) in the decentralized clinic model. The difference in the observed and expected incidence rate was 4.3 (95% CI: 0.44–39.57) for the decentralized clinic model.

## Conclusion

Although incidence of HIV seroconversion was highest in the decentralized clinic model, this difference may be due to the higher sexual risk behavior among study participants in the decentralized model rather than the type of service delivery. The study findings imply that any of the models can effectively deliver PrEP services.

## Introduction

After almost 40 years of the HIV epidemic, novel and effective HIV prevention strategies, especially for high-risk populations, are still urgently needed. Implementation of strategies at the population level that have been proven effective in clinical trials is critical [1]. Also, lowering the costs by targeting delivery to populations at the highest risk for HIV-1 is essential. In the past 10 years interest in antiretroviral-based strategies for prevention of sexual HIV-1 transmission has increased, and antiretroviral-based HIV-1 prevention interventions are among the most promising strategies for reducing the spread of HIV-1 [2]. Antiretrovirals have the potential to prevent HIV-1 infection by 1) antiretroviral therapy (ART) to reduce the infectiousness of HIV-1-infected persons, also known as treatment as prevention (TasP); and 2) oral or topical pre-exposure prophylaxis (PrEP) for uninfected persons who have repeated and ongoing HIV-1 exposure [3].

Studies have demonstrated that PrEP has the potential to reduce HIV transmission on a population level [2, 3]. The next step is determining the best methods for implementing safe and effective PrEP in various settings, including through demonstration projects. This step will help translate research findings into practice by evaluating PrEP use outside of clinical trial settings in anticipation of the roll-out of these strategies in each country. Demonstration projects will assess the realities of PrEP adoption, including requirements for regular HIV testing, ongoing medical monitoring, and assessment of side effects and toxicities of the medications [4].

The first of many large randomized efficacy trials was the iPrEx open-label extension study conducted in 11 sites in the United States, Brazil, Peru and Ecuador. It was designed to provide information about the safety and efficacy of PrEP and the behavior of people taking PrEP over the long term [5]. Since conclusion of that study, PrEP demonstration projects have been implemented for various populations [6, 7] in Africa [8, 9], Asia [10], America [11], Australia [12], South America [13] and Europe [14].

However, few studies on HIV-1 serodiscordant couples have been conducted. The Partners' clinical trials on PrEP use by HIV serodiscordant couples reported a 67% relative reduction in HIV-1 incidence with single-drug tenofovir disoproxil fumarate (TDF) prophylaxis and 75% reduction with emtricitabine (FTC)/TDF combination (Truvada), with no significant difference in PrEP efficacy by drug type or by sex in both the primary efficacy study [15] and the open-label extension study [16]. An implementation study conducted in Kenya and Uganda

by the Partners' study group [17] to determine the effectiveness of PrEP in service delivery settings found that PrEP access and use resulted in 93% reduction in HIV-1 incidence in uninfected female partners and a 100% reduction in uninfected male partners.

The Nigeria PrEP open-label demonstration study [18] recruited HIV-1 serodiscordant couples and was designed to determine the effectiveness of PrEP roll-out models. This strategy was needed because of the challenges of translating research findings into health service programs, especially where health care delivery systems are weak. This problem is evident with the programs for the prevention of mother-to-child transmission in Nigeria; despite the efficacy of the program, only about 44% of HIV-positive pregnant women had access to it in 2018 [19]. The Nigeria PrEP demonstration study is an opportunity to answer critical questions about how best to incorporate PrEP into routine health services for serodiscordant couples and how to make PrEP accessible to HIV-negative men and women in serodiscordant relationships. Serodiscordant couples are identified as priority populations for PrEP [20].

The aim of the Nigeria PrEP demonstration project was to evaluate the effectiveness of three models for service delivery as part of a combination HIV prevention strategy [18, 21]. The models, which were developed through an extensive consultative process, would deliver PrEP for the HIV-negative partner in a serodiscordant relationship and treatment for the HIV-positive partner. The primary measure was the number of new infections prevented by PrEP.

## Methods

### Study population

The study recruited HIV-1 serodiscordant sexually active (defined as having had vaginal intercourse at least 6 times in the previous three months) heterosexual couples willing to participate in the study as a couple and intending to remain as a couple for the next 12 months. The couple would also have not taken PrEP as medication before enrolment in the study.

At study enrolment, uninfected partners were required to have adequate renal function (defined as normal creatinine levels of >130 mmol/l and estimated creatinine clearance rate of >60ml/min). Pregnant or breast-feeding HIV-1 uninfected female partners were excluded from the study. HIV-1-infected partners (index participants) were diagnosed according to the national HIV testing algorithm [21]. All recruited HIV-1-infected partners had no history of World Health Organization stage III or IV conditions and were not virologically suppressed at the time of enrolment in the study. Current pregnancy or breastfeeding were not exclusion criteria for infected partners.

### Sample size

A sample size of 390 was computed for the study with 130 couples per model. The sample size was based on the formula of comparing two or more proportions, with a minimum retention rate of 75% [22, 23] for each study site, with an assumption of a 15% difference across the sites at a two-tailed significance level of 5% and power of 80% and adjusting for a 15% loss to follow up.

### Study sites

The study was conducted at ART sites supported by FHI360/SIDHAS or the Harvard/PEPFAR projects. The sites were the Nnamdi Azikwe University Teaching Hospital at Nnewi in Anambra State in South-East Nigeria, the University Teaching Hospital in Calabar located in South-South Nigeria, Cross Rivers State and the University Teaching Hospital at Jos, Plateau State in

North-Central. The sites were tertiary healthcare institutions located in distinct geographical locations, chosen to avoid contamination and diffusion bias. The study was also conducted at sites where access to quality ART services for eligible HIV positive partners could be assured.

## Program description

The project implemented three models of PrEP delivery: outpatient, decentralized and ART clinic models. An observational cohort study design was adopted in delivering PrEP by the three models in parallel. Each site implemented PrEP either as an outpatient clinic model or an ART clinic model or a decentralized care model. Each model had equal allocation of participants. The sites are located in three states, geographically distant from one another, to limit contamination and information diffusion bias.

**Outpatient clinic model.** Outpatient services are widely available in Nigeria. An outpatient model integrated HIV prevention services, including PrEP, into routine general outpatient care. This model was implemented at the University Teaching Hospital at Calabar, Cross Rivers State.

**ART clinic model.** ART clinics have trained specialized care providers who were prescribing ART and PrEP to clients taking PrEP. This model was implemented at the University Teaching Hospital at Nnewi, Anambra State.

**Decentralized care model.** The decentralized care model facilitates access to routine ART care for community members; it links clients who are receiving ART to specialized care at a central tertiary hospital. The decentralized care model linked six primary and secondary care centres to a tertiary care centre. This model was implemented by the University Teaching Hospital at Jos, Plateau State.

## Recruitment

Each site developed its own local recruitment and screening methods. The methods had protocol-specified requirements for eligibility tailored to be most efficient for the local study setting and target study population. The local recruitment and screening methods were designed according to the outcome of the PrEP formative study [21]. Recruitment strategies included partnering with existing HIV counseling and testing (HCT) centers and with civil society organizations working with families and couples, public promotion of HCT for couples by community organizations such as churches and mosques, and community mobilization around couples HCT promotion (during Valentine's Day and World AIDS Day).

Informed consent for screening was obtained individually from each partner. The screening process proceeded in a stepwise manner for both partners until either all screening procedures were completed or one or both of the partners were determined to be ineligible. Although all required screening procedures might be completed in as few as two visits for each partner, additional visits could be conducted as needed (for example, if one or both partners want more time to consider whether to enroll in the study). Couples together attended at least one screening visit, and at least one couple counseling session took place during the screening process. Couples eligible for the study gave written consent for participation and enrolment. Each partner provided independent informed consent for study participation.

## Study procedure

We adopted the Partner's PrEP study approach as proposed by the World Health Organization [24]. Study duration was 30 months, with accrual conducted for 12 months and follow-up continuing for 18 months after accrual. Potential study participants were screened for eligibility, and eligible participants were enrolled in the study within 30 days of screening. Visits took

place at screening and enrolment, one month after enrolment, and quarterly thereafter for up to 12 months. The first couple were recruited 10th November, 2015 and last follow-up was 11th June, 2018.

At enrolment, couples were counseled on the benefits of immediate enrolment for ART and PrEP access. All HIV-1-infected partners were advised to initiate ART according to national treatment guidelines. At the time of the study, the guidelines included all HIV-1-infected partners in HIV-1-serodiscordant relationships, regardless of CD4 count. Partners were offered the nationally recommended ART regimens (preferred regimen: TDF, lamivudine, and efavirenz, with zidovudine and nevirapine as alternative agents).

All HIV-1-infected partners initiated ART at the time of couple enrolment, in line with the national ART guidelines for management of HIV serodiscordant partners [21]. No patient with a plasma HIV-1 RNA <400 copies/mL was enrolled in the study. HIV-1-uninfected partners were given PrEP (combination FTC/TDF 200/300 mg once daily) at the study sites, as PrEP was not available otherwise during the study. PrEP was provided until the HIV-1-infected partner had become virologically suppressed or for as long as the participants wanted, even after the infected partner had become suppressed.

Throughout the study PrEP medication was supplied free to participants. Study products were securely stored at controlled room temperature below 30˚C until administered. PrEP was dispensed by designated individuals in quantities expected to be sufficient until the participant's next visit, which was usually 90 days. If participants required supplies between visits, they were instructed to contact the study clinic to request them. Study participants were asked to bring their drug bottles at each visit. All returned drugs were reconciled with the number dispensed, and the number was logged.

At the screening stage, demographic and behavioral information was collected, along with laboratory results to establish participant eligibility. For HIV-1-uninfected partners, the tests included serum creatinine and hepatitis B surface antigen; for HIV-1-infected partners, the tests included CD4 count and plasma HIV-1 viral load; for both partners, HIV-1 rapid testing was conducted according to national algorithms. At enrolment, HIV-1 testing was performed for HIV-1-uninfected partners to confirm eligibility (HIV-1 seronegative at the time of study start). Couples were counseled about ART and PrEP. HIV-1-uninfected partners were offered PrEP; HIV-1-infected partners were counseled about ART guidelines and started on ART if they were eligible and interested at the time of enrolment of their HIV-1-uninfected partners on PrEP.

Eligible clients were counseled on adherence, HIV risk reduction and contraception. Adherence of HIV-1-uninfected participants was measured by self-report and pill counts. In addition, the Medication Electronic Monitoring System (MEMS) cap, manufactured by AAR-DEX Group, was used to complement data generated in the clinic on self-reported pill use for 113 (32.6%) of clients. The MEMS cap generated information on the daily consumption of medication by electronically capturing the frequency that drug bottles were opened. Outcomes of the adherence profile generated from self-reported pill counts and the MEMS cap were used to counsel patients on how to improve or adhere to drug use. At 12 months, a blood sample of 10% of the study sample was assayed for the concentration of tenofovir.

At enrolment, HIV-1-uninfected women had a pregnancy test and were excluded from the study if pregnant. Women were asked about pregnancy at every scheduled visit, and pregnancy testing was repeated when indicated. PrEP was continued during pregnancy and breastfeeding.

Every 6 months, in addition to regular quarterly visit procedures, CD4 counts and plasma viral load were assessed for HIV-1-infected partners. Laboratory safety was assessed at enrolment, month 12 and study exit. At these visits, blood samples were collected for measurement

of blood urea nitorgen, electrolytes, creatinine, liver function tests, full blood count, calcium and phosphate values. When the HIV-1-infected partner had used ART for 6 months and was virologically suppressed, is continuation of PrEP for the HIV-1-uninfected partner was advised after counseling the couple. For participants who were initially uninfected but seroconverted during the study, PrEP was discontinued. A blood specimen was taken and drug resistance was profiled.

## Study site preparation

As part of the study intervention, a Quality Improvement (QI) approach was used to assist the service delivery outlets in expanding their services to include PrEP and TasP. A study internal monitor, experienced with QI processes, worked with service delivery outlets to analyze existing services before study participants were enrolled. External and internal ideas were considered for improving the quality of service in family planning; counseling and provision of contraception; counseling and treatment of sexually transmitted infections and HIV; and general clinic processes (clinic flow, documentation, and follow-up).

The advisor conducted two monitoring exercises and a site-initiation assessment before activating the site to ensure that procedures were in place for the study, including the dispensing of PrEP and TasP. Also, each PrEP and TasP delivery site was monitored at half-yearly with goals and timelines agreed on for monitoring of outcomes. A two-step approach for strengthening the services and introducing PrEP and TasP helped create a cadre of service providers who could maintain the quality of services and cope with unexpected or unanticipated situations, in contrast to a traditional prescriptive, top-down approach of service delivery.

## Data analysis

The primary effectiveness endpoint for the study was incident HIV-1 infection, defined as seroconversion, with exclusion of patients found to be HIV-1-infected before study initiation. Descriptive statistics were used to summarize the socio-demographic and biological characteristics of the study participants, including sexual risk behaviours of the HIV-1-infected partners at enrolment. Normally distributed continuous data were summarized by the use of mean and standard deviation; non-normally distributed continuous data were summarised with a median and interquartile range. Discrete variables were summarized using frequencies and percentages. All summary data were disaggregated by PrEP service delivery model.

Survival analysis was conducted with Kaplan-Meier Survival function to estimate the median retention time (months) of the study participants in each of the study models, and log-rank test for equality of survival functions was conducted to test for significant differences among the three models. The event of interest was the discontinuation of couple's visit. Those who discontinued before 12 months were recorded to have an event at the time of discontinuation. Those whose period of stay was longer than 12 months were censored at 12 months. The Cox regression model was used to examine the influence of multivariate variables and confounding variables across the three models, and to do pairwise comparisons of the models Bonferroni correction was applied to the original $\alpha$ (0.05) to obtain a new $\alpha$ (0.01667).

Basic statistical tests (analysis of variance and chi-square test) were conducted to test for a significant difference in mean, median and proportions of the characteristics of the study participants across the three models at enrolment. For HIV-1-infected partners, the CD4 count and viral load status at enrolment were used to characterize the cohort and describe the uptake of and adherence to ART and PrEP.

The socioeconomic characteristics of each study participant was assessed by combining 10 socioeconomic variables on housing and ownership of fixed assets into a measure of household

wealth. These items were electricity, television, refrigerator, running water, concrete floor, mattress, car, mobile phone, "number of rooms in a house" and household size. All the variables were binary variables with 0 (No) or 1 (Yes) response, except the "number of rooms in a house" and household size, which were integer variables. Principal Component Analysis was applied to the data to derive socioeconomic scores (SES). The first principal component extracted with an associated eigenvalue of 2.05 was taken as the measure of the SES. The first principal component scores were first categorized into quintiles and later re-categorized into low (1st & 2nd Quintiles), middle (3rd Quintiles) and high (4th & 5th Quintiles) SES.

The HIV-1 positive incidence before initiation of the study was compared with the HIV-1 incidence at the end of the study per model. The pre-study initiation data at the study site was collected 12 months prior to study initiation. HIV-1-positive partners in a serodiscordant relationship were started on ART immediately after diagnosis, and they had access to continued ART consistent with the national ART policy [22]. The HIV-1-negative partner in the serodiscordant relationship was counseled on the use of condoms. The number of HIV seroconversions prior to study initiation was computed. The incidence rate ratio was computed comparing HIV-1 incidence in the present study to the mean site incidence; a 95% confidence interval was calculated using a Poisson distribution, and the *p*-value was computed. Models of the HIV-1-uninfected partner were also constructed according to sex and enrolment plasma HIV-1 RNA concentration of the HIV-1-infected partner to create estimates for each of these subgroups. Analyses were conducted with Stata/SE 14.0 for Windows.

### Ethics clearance

The study protocol was approved 10th January, 2014 by the Ethics Committee of the National Institute of Medical Research, Lagos (IRB-14-254). The Ethics Committee of the University of Jos Teaching Hospital (DCS/ADM/127/XiX/6011), University of Calabar Teaching Hospitals (UCTH/HREC/33/490) and the Nnamdi Azikwe University Teaching Hospital, Nnewi (NAUTH/CS/66/Vol.7/21) also gave study approval. All participants provided written consent.

Study participants received no financial compensation. On the strength of the ethical approvals obtained from the Nigerian Institute of Medical Research Ethics Committee and the individual institutional IRBs the study commenced. The process of obtaining trial registration of PrEP intervention models delayed due to administrative reasons. The authors confirm that all ongoing and related trials for this drug/intervention are registered.

### Results

Of the 544 couples screened 237 (43.6%) were not eligible for enrolment for reasons ranging from low HIV viral load in HIV-1-infected partners and low creatinine clearance/high creatinine levels in HIV-1-uninfected partners. Of the 307 couples eligible for enrolment, 10 (3.3%) were not enrolled due to lack of interest in study participation and failure to return to the clinic for study enrolment. A total of 297 (96.7%) of the eligible HIV-1-serodiscordant couples were enrolled in the PrEP demonstration study in Nigeria. Enrollment into the three PrEP models had 130 serodiscordant couples in ART clinic model, 94 couples in the out-patient clinic model and 73 couples in the decentralized clinic model. Fig 1 is the flowchart of the screening, enrolment, and follow-up data for the study.

### Study participants' profile

Table 1 highlights the socio-demographic profile of the study participants across the three PrEP models: The mean ages of un-infected partners were 40.4 ± 8.0 years, 37.4 ± 9.1 years

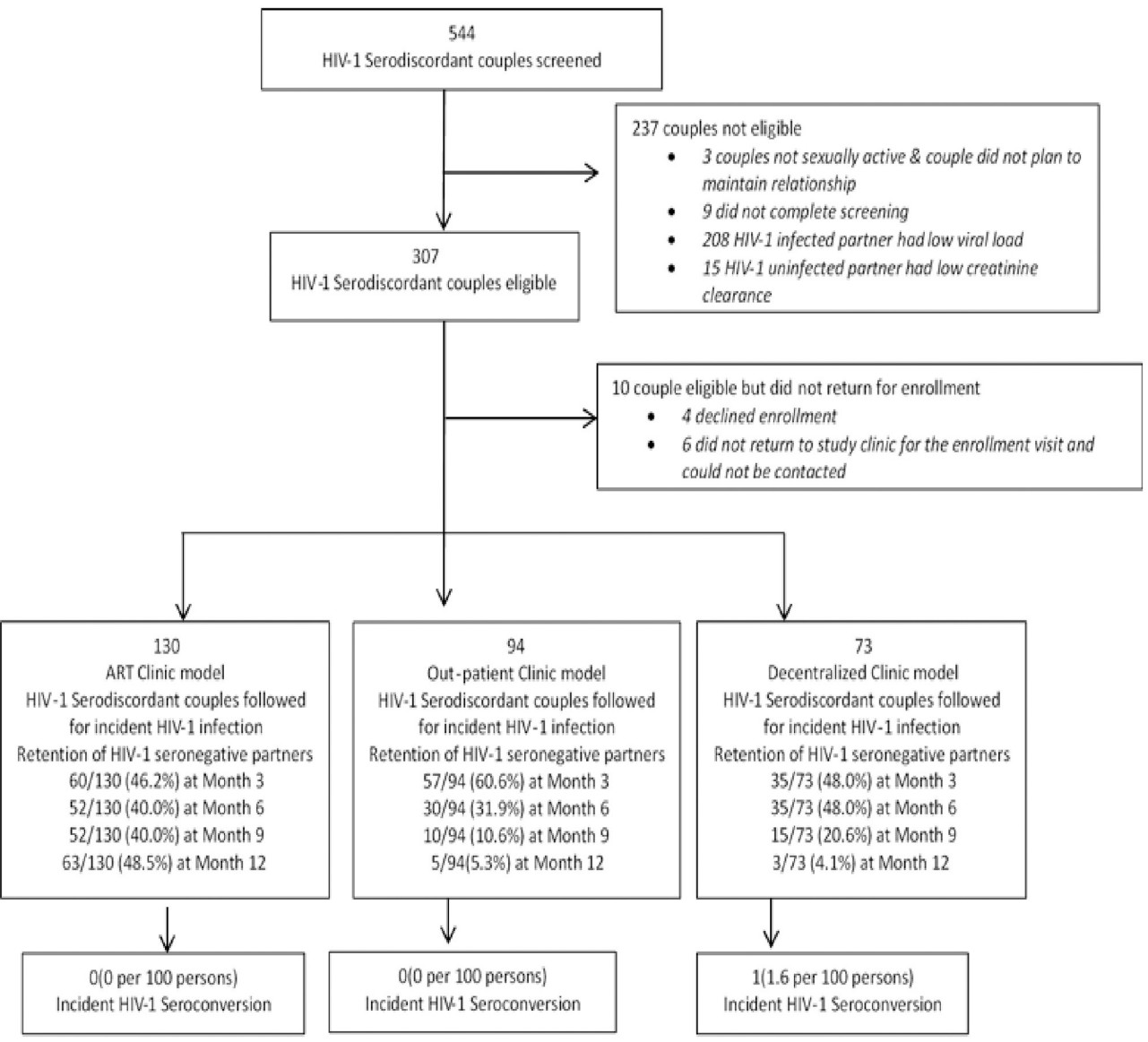

**Fig 1. Screening, enrollment and follow up flow chart.**

and 38.9 ± 8.6 years respectively in the ART clinic model, out-patient clinic model and decentralized clinic model. There were more uninfected females partners in the ART clinic model (69/130, 53.1%) and more uninfected males in the out-patient clinic model (58/94, 61.7%) and decentralized clinic model (48/73, 65.8%).

The wealth status of couples by model revealed 38.0% of couples in the ART clinic model, 37.2% of couples in the out-patient clinic model and 50.7% of couples in the decentralized clinic models had low wealth status. Participants who received PrEP in the ART-clinic model and their partners were older than those who received PrEP in the outpatient and decentralized models; more males than females received PrEP in the outpatient clinic model and the decentralized clinic model; and most couples were of the middle wealth quantile. The majority of the couples also had completed secondary education.

**Table 1. Baseline socio-demographic, biological and sexual risk behavior of participants by PrEP model (N = 297).**

| | ART Clinic model (N = 130) | | Out-patient Clinic model (N = 94) | | Decentralized Clinic model (N = 73) | |
|---|---|---|---|---|---|---|
| | HIV-1 infected partners | HIV-uninfected partners | HIV-1 infected partners | HIV-uninfected partners | HIV-1 infected partners | HIV-uninfected partners |
| **Age (years)** Mean (SD) | 40.7(8.9) | 40.4(8.0) | 34.9(8.2) | 37.4 (9.1) | 34.9(8.2) | 38.9 (8.6) |
| **Age (years)** | | | | | | |
| *Less than or equal to 24* | 7(5.4%) | 1 (0.8%) | 8(8.5%) | 5 (5.3%) | 7(9.6%) | 1 (1.4%) |
| *Greater than or equal to 25* | 123(94.6%) | 129 (99.2%) | 86(91.5%) | 89 (94.7%) | 66(90.4%) | 72 (98.6%) |
| **Sex** | | | | | | |
| *Male* | 71(54.6%) | 61(46.9%) | 35(37.2%) | 58(61.7%) | 21(28.8%) | 48(65.8%) |
| *Female* | 59(45.4%) | 69 (53.1%) | 59(62.8%) | 36(38.3%) | 52(71.2%) | 25 (34.3%) |
| **Wealth status** | | | | | | |
| *Low* | 49(38.0%) | | 35(37.2%) | | 37(50.7%) | |
| *Medium* | 43(33.3%) | | 42(44.7%) | | 21(28.8%) | |
| *High* | 37(28.7%) | | 17(18.1%) | | 15(20.6%) | |
| **Educational status** | | | | | | |
| *None* | 2 (1.5%) | 1(0.8%) | 4 (4.3%) | 1(1.1%) | 5 (6.9%) | 1 (1.4%) |
| *Primary* | 39 (30.0%) | 27(20.8%) | 17 (18.1%) | 14(14.9%) | 14(19.2%) | 10 (13.7%) |
| *Secondary* | 70 (53.9%) | 70(53.9%) | 44 (46.8) | 36(38.3%) | 34(46.6%) | 28(38.4%) |
| *Higher* | 19 (14.6%) | 26(20.0%) | 29(30.9%) | 37(39.4%) | 20(27.4%) | 32(43.8%) |
| *Missing* | 1(1%) | 6 (4.6%) | 5(5%) | 6 (6.4%) | 1(1%) | 2 (2.7%) |
| **Biological parameters** | | | | | | |
| HIV-1 plasma RNA $\geq$50,000 viral load copies/mL Median (IQR) | 47,327.5 (5,397–175,374) | | 29,933.5 (2,284–155,619) | | 63,345.5 (8,734–302,941.5) | |
| CD4 count/μL Median (IQR) | 247 (123–359) | | 244 (106–424) | | 246 (107–401) | |
| **Sexual risk behavior** | | | | | | |
| Number of sex acts, prior month | 3 (1–4) | 2 [1–4] | 3 (1–4) | 2 [1–3] | 2 (1–5) | 3 [2–5] |
| Unprotected sex acts, prior month | 38 (29.2%) | 52 (40.0%) | 25(26.6%) | 47 (50.0%) | 36(49.3%) | 39 (53.4%) |
| Sex with outside partner, prior month | 1 (0.8%) | 1 (0.8%) | 1(1.1%) | 3 (3.2%) | 7(9.6%) | 3 (4.1%) |

## Viral load and a CD4 count of the HIV-infected partners

Table 1 highlights the median CD4 count and viral load of the HIV-1-infected partners recruited for the study. The median CD4 counts were 247, 244 and 246 among HIV-1 infected partners in the ART clinic model, out-patients clinic model and decentralized clinic model respectively. However, median viral load among HIV-infected partners varied across the three models 47,327.5 ART clinic model, 29,933.5 out-patient model and 63.345.5 decentralized model.

## Sexual risk behavior

The number of sex acts per study participants in the month prior to enrolment were similar per study model (Table 1). However, the number of unprotected sex acts in the prior month was least at the outpatient clinic model, and the number of participants who had sex outside the relationship in the prior month to enrolment was highest in the decentralized clinic model.

**Table 2. Retention data at 3, 6, 9 and 12 months by PrEP model.**

| Variables | HIV-1 Sero-discordant couples | | |
|---|---|---|---|
| | **ART Clinic model** | **Out-patient Clinic model** | **Decentralized Clinic model** |
| **Screening statistics** | | | |
| Number of couple screened | 232 | 142 | 170 |
| **Retention statistics** | | | |
| Number of couple enrolled | 130 | 94 | 73 |
| Number (%) of couple visit at month 3 | 84 (64.6%) | 62 (66.0%) | 58 (79.5%) |
| Number (%) of couple visits at month 6 | 72 (55.4%) | 50 (53.2%) | 60 (82.2%) |
| Number (%) of couple visits at month 9 | 67 (51.5%) | 41 (43.6%) | 54 (74.05) |
| Number (%) of couple visits at month 12 | 83 (63.8%) | 79 (84.0%) | 59 (80.8%) |
| Median retention months (50%) | 10 | 11 | 12 |

## Retention rates

Table 2 describes the screening, enrolment, and retention () data for the HIV-infected and HIV-uninfected partners per model at 3, 6, 9 and 12 months. A total of 225 person-years follow-up were accrued for the couples, with a median follow-up of 11 months per couple for assessment of incident HIV-1 infection (IQR 5–12). Retention of HIV-1 uninfected partners for assessment of HIV-1 acquisition decreased from month 3 to month 9 follow-up period. As shown in Fig 2, the log-rank test for equality-of-survival functions revealed no significant difference (p>0.05) in the couple retention rates in the three models. However, retention of the couples decreased from month 3 to month 9.

## Adherence

All HIV-1-uninfected partners initiated PrEP at enrolment. At months 3, 6, 9 and 12 follow-up visits, 152 (74.5%), 117 (64.3%), 77 (47.5%) and 71 (32.1%) subjects continued to receive PrEP, respectively. Adherence to PrEP, as measured by self-report, indicated that drug adherence was consistent for the HIV-1-infected partners over the study period, whereas it decreased over time for the HIV-1-uninfected partner on PrEP (see Fig 3). When analyzed per model, differences for both HIV-1 infected and HIV-1 uninfected partners' drug intake profile were evident. At months 3, 6 and 9, the decentralized model had the highest proportion of HIV-1 infected partners receiving ART, whereas at months 9 and 12 the outpatient model had the highest proportion of HIV-1-uninfected partners receiving PrEP (p<0.001).

By MEMS cap measurement the percentage of days with PrEP intake during the study implementation ranged from 17.6 to 100, with a median of 91.7% and an average of 85% (IQR: 83.5–96.9), as shown in Fig 3. In the selected sample of individuals receiving PrEP ($n$ = 50 participants at month 12 study visit), tenofovir was detected in plasma in 85% of samples (372/438).

## Incidence of HIV-1 infection

Fig 3 shows the incidence of HIV at each model prior to the commencement of the study and at the end of the study. Prior to the commencement of the study, the outpatients model (Calabar) recorded one seroconversion in the 199 couples followed for one year. The ART clinic model (Nnewi) had two seroconversions from the 435 couples followed for one year. The decentralized model reported one seroconversion among the 260 couples followed for a year. While the observed HIV incidence per 100 person-years was zero in the general outpatient

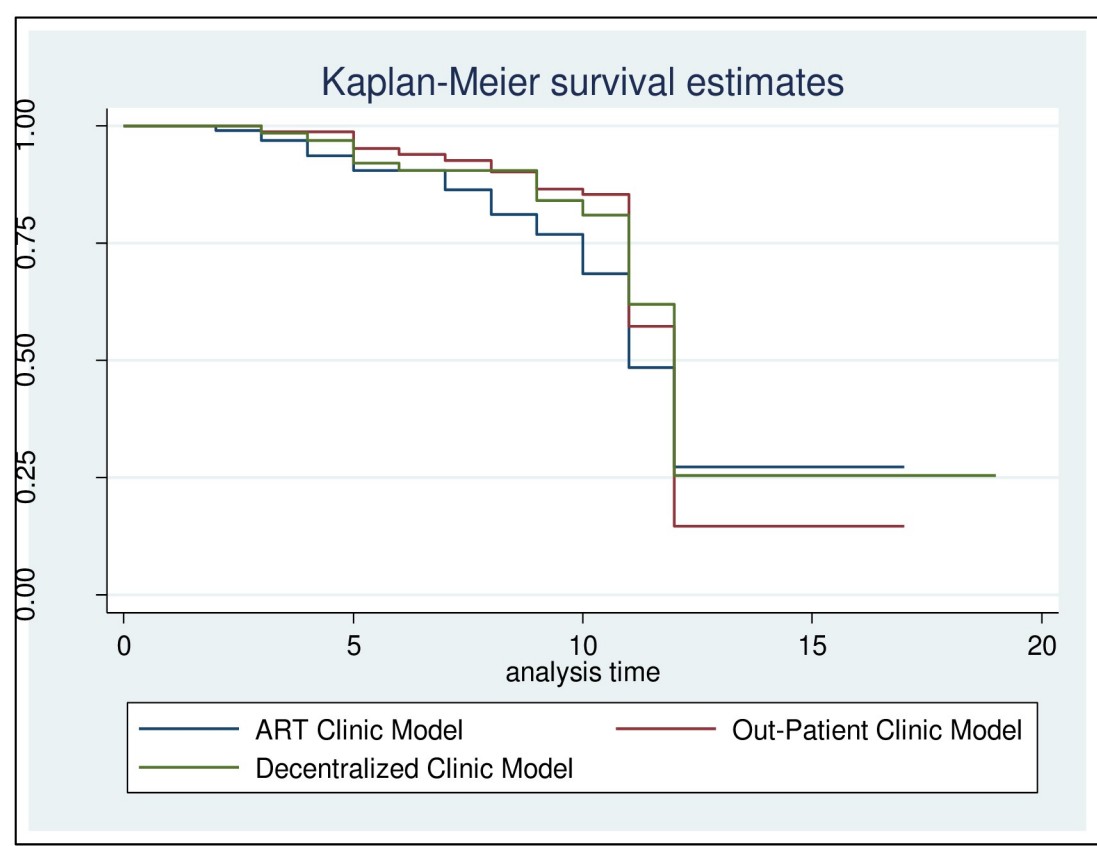

**Fig 2. Retention rate of the couples by the three models.** Log-rank trend test of equality of survivor functions, x2 = 0.75, p-value = 0.3852.

clinic and ART clinic models, it was 1.6 (95% CI: 0.04–9.1) in the decentralized clinic model. The difference in the observed and expected incidence rate was 4.3 (95% CI: 0.44–39.57) for the decentralized clinic model and zero for the general outpatient model and ART clinic model.

The influence of multiple factors that may affect seroconversion among HIV-1 uninfected partners were compared across the three PrEP delivery models using the Cox regression multivariate analyses. The results are presented in Tables 3 & 4 below. There were no statistically significant difference in the influence of age, sex, education and wealth status. However, having sex with an outside partner in prior month among HIV-1 uninfected partner was statistically significant in the out-patient PrEP clinic model had a hazard ratio of 23.61 95% confidence interval of (3.87–144.02). This risky sexual behavior was not found to confer significant risk in the ART clinic model hazard ratio 0.68 95% confidence interval (0.08–5.41) and in the decentralized clinic model hazard ratio 0.90 with 95% confidence interval (0.11–7.35).

## Discussion

Results of the study indicate that although the rate of retention of study participants on PrEP did not differ by model, adherence to PrEP differed according to model: at months 9 and 12, the outpatient model had the highest rate of PrEP clients retained in care. The higher retention rate when PrEP is integrated into the general outpatient delivery system may reflect less stigma

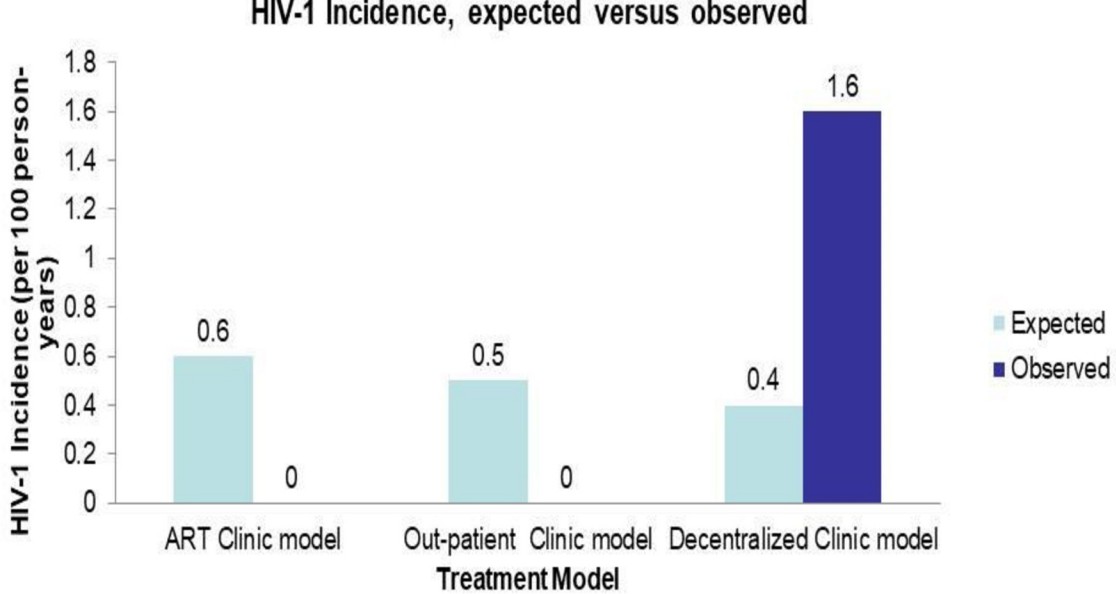

**Fig 3. HIV-1 Incidence expected versus observed by PrEP model.**

| Study model | Expected # HIV infections | Expected HIV-1 Incidence, per 100 person years (95%CI) | Observe # HIV infections | Observed HIV-1 Incidence, per 100 person years (95%CI) | Incidence Rate Ratio (IRR) Observed vs. Expected (95%CI) | P value |
|---|---|---|---|---|---|---|
| ART Clinic model | 2 | 0.6 (0.07—2.1) | 0 | 0.0 (0.0—0.43) | 0 (0.0—7.59) | >0.05 |
| Out-patient Clinic model | 1 | 0.5 (0.01—2.8) | 0 | 0.0 (0.0—4.8) | 0 (0.0—10.36) | >0.05 |
| Decentralized Clinic model | 1 | 0.4 (0.01—2.1) | 1 | 1.6 (0.04—9.1) | 4.2 (0.44—39.57) | >0.05 |

of HIV treatment than when it is provided in specialized HIV clinics, as in ART clinics and decentralized service models. Stigma associated with ART is a deterrent for HIV care [25], so PrEP provided through outlets that also provide ART services may reduce retention of PrEP clients, as suggested by these study results.

There was a single seroconversion in the decentralized service delivery model. At the site implementing the decentralized service delivery model, the number of sex acts in the prior month and the number of participants who had other sex partners was significantly higher than in participants in the other two PrEP clinic models. This difference may indicate that understanding the typology of sexual risk behavior of persons who take PrEP could be used to improve risk-reduction counseling at the clinics where PrEP is being provided. However, this

**Table 3. Cox regression analysis comparing influence of factors stratified by models.**

| | ART Clinic Model | | | | Out-patient Clinic Model | | | | Decentralized Clinic Model | | | |
|---|---|---|---|---|---|---|---|---|---|---|---|---|
| | HR | [95% CI] | | p value | HR | [95% CI] | | p value | HR | [95% CI] | | p value |
| **Age of HIV-1 infected partner** | | | | | | | | | | | | |
| *Less than or equal to 24 years* | 1.00 | | | | 1.00 | | | | 1.00 | | | |
| *Greater than or equal to 25 years* | 0.75 | 0.23 | 2.46 | 0.631 | 1.55 | 0.56 | 4.26 | 0.399 | 1.17 | 0.41 | 3.37 | 0.766 |
| **Age of HIV-uninfected partner** | | | | | | | | | | | | |
| *Less than or equal to 24 years* | 1.00 | | | | 1.00 | | | | 1.00 | | | |
| *Greater than or equal to 25 years* | 0.52 | 0.06 | 4.33 | 0.543 | 0.35 | 0.11 | 1.15 | 0.083 | 1.00 | | | |
| **Sex of HIV-1 uninfected partner** | | | | | | | | | | | | |
| *Male* | 1.00 | | | | 1.00 | | | | 1.00 | | | |
| *Female* | 2.34 | 0.40 | 13.65 | 0.344 | 0.97 | 0.12 | 8.22 | 0.981 | 3.30 | 0.96 | 11.36 | 0.058 |
| **Sex of HIV-1 infected partner** | | | | | | | | | | | | |
| *Male* | 1.00 | | | | 1.00 | | | | 1.00 | | | |
| Female | 1.88 | 0.32 | 11.06 | 0.485 | 0.63 | 0.07 | 5.54 | 0.680 | 2.79 | 0.74 | 10.46 | 0.129 |
| **Wealth index** | | | | | | | | | | | | |
| *Low* | 1.00 | | | | 1.00 | | | | 1.00 | | | |
| *Middle* | 0.94 | 0.47 | 1.88 | 0.858 | 1.37 | 0.76 | 2.49 | 0.297 | 0.70 | 0.32 | 1.56 | 0.386 |
| *High* | 0.76 | 0.39 | 1.46 | 0.406 | 0.77 | 0.35 | 1.69 | 0.510 | 2.00 | 0.79 | 5.06 | 0.144 |
| **Educational level of HIV-1 infected partner** | | | | | | | | | | | | |
| None/Primary | 1.00 | | | | 1.00 | | | | 1.00 | | | |
| Secondary | 0.89 | 0.51 | 1.53 | 0.672 | 1.49 | 0.68 | 3.24 | 0.315 | 0.95 | 0.42 | 2.17 | 0.904 |
| Higher | 0.87 | 0.34 | 2.22 | 0.764 | 1.83 | 0.69 | 4.88 | 0.224 | 0.51 | 0.18 | 1.43 | 0.200 |
| **Educational level of HIV-1 uninfected partner** | | | | | | | | | | | | |
| None/Primary | 1.00 | | | | 1.00 | | | | 1.00 | | | |
| Secondary | 0.68 | 0.33 | 1.39 | 0.286 | 0.86 | 0.36 | 2.02 | 0.726 | 1.03 | 0.36 | 2.93 | 0.961 |
| Higher | 0.82 | 0.33 | 2.04 | 0.672 | 0.52 | 0.20 | 1.38 | 0.189 | 1.88 | 0.63 | 5.59 | 0.258 |
| Missing | 0.50 | 0.11 | 2.34 | 0.378 | 1.63 | 0.30 | 9.00 | 0.573 | 0.38 | 0.04 | 3.85 | 0.411 |
| **Number of sex acts in prior month among HIV-1 infected partner** | 1.08 | 0.95 | 1.23 | 0.239 | 1.01 | 0.93 | 1.11 | 0.758 | 1.08 | 1.00 | 1.17 | 0.050 |
| **Number of sex acts in prior month among HIV-1 uninfected partner** | 0.95 | 0.83 | 1.07 | 0.383 | 1.06 | 0.97 | 1.17 | 0.204 | 1.02 | 0.94 | 1.11 | 0.581 |
| **Unprotected sex acts in prior month among HIV-1 uninfected partner's having** | | | | | | | | | | | | |
| No | 1.00 | | | | 1.00 | | | | 1.00 | | | |
| Yes | 1.42 | 0.77 | 2.63 | 0.264 | 1.37 | 0.76 | 2.47 | 0.300 | 1.51 | 0.62 | 3.67 | 0.364 |
| **Unprotected sex acts in prior month among HIV-1 infected partner** | | | | | | | | | | | | |
| No | 1.00 | | | | 1.00 | | | | 1.00 | | | |
| Yes | 1.25 | 0.68 | 2.29 | 0.475 | 1.38 | 0.80 | 2.36 | 0.249 | 0.91 | 0.41 | 1.99 | 0.809 |
| **Sex with outside partner in prior month among HIV-1 uninfected partner** | | | | | | | | | | | | |
| No | 1.00 | | | | 1.00 | | | | 1.00 | | | |
| Yes | 0.68 | 0.08 | 5.41 | 0.712 | 23.61 | 3.87 | 144.02 | 0.001 | 0.90 | 0.11 | 7.35 | 0.920 |
| **Sex with outside partner in prior month among HIV-1 infected partner** | | | | | | | | | | | | |
| No | 1.00 | | | | 1.00 | | | | 1.00 | | | |
| Yes | 1.27 | 0.14 | 11.60 | 0.833 | 1.00 | | | | 0.95 | 0.35 | 2.56 | 0.915 |

interpretation should be made with caution, as this was a single seroconversion, and the incidence had wide confidence intervals. The individual had a history of non-adherence to PrEP, and the seroconversion occurred at the time the HIV-infected partner was virologically suppressed; there also may have been infidelity by the HIV-uninfected partner, but the source of infection could not be confirmed because the study did not conduct HIV genotyping.

**Table 4. Cox regression analyses comparing the models.**

|  | Adjusted HR | 95% CI | | p-value |
|---|---|---|---|---|
| ART Clinic Model* vs. Out-patient Clinic Model | 0.94 | 0.59 | 1.50 | 0.789 |
| ART Clinic Model* vs. Decentralized Clinic Model | 0.79 | 0.44 | 1.40 | 0.412 |
| Out-patient Clinic Model* vs. Decentralized Clinic Model | 0.94 | 0.57 | 1.55 | 0.807 |

*Reference category

Adjusted for *Age of HIV-1 infected partner, Age of HIV-uninfected partner, Sex of HIV-1 uninfected partner, Sex of HIV-1 infected partner, Wealth index, Educational level of HIV-1 infected partner, Educational level of HIV-1 uninfected partner, Number of sex acts in prior month among HIV-1 infected partner, Number of sex acts in prior month among HIV-1 uninfected partner, Unprotected sex acts in prior month among HIV-1 uninfected partner's having, Unprotected sex acts in prior month among HIV-1 infected partner, Sex with outside partner in prior month among HIV-1 uninfected partner, Sex with outside partner in prior month among HIV-1 infected partner.*
Bonferroni Correction applied; α (p-value) set at 0.01667.

The study also showed differences in the ART adherence profile of HIV-1 infected and HIV-1 uninfected partners: adherence was stable for the HIV-positive partner, whereas it decreased over time for the HIV-negative partner. This difference is disappointing, as one of the objectives of the enrolment of couples was continued adherence by both partners. The finding suggests, however, that HIV-positive partners in HIV-serodiscordant relationships do not always provide good adherence support for HIV-negative partners. Although evidence suggests that partner support enhances the uptake of ART for persons who are HIV-positive [25–27], there is less evidence that partner support improves adherence to PrEP [28].

Some clients found adherence to PrEP challenging for numerous reasons. One reason was concern about possible drug side effects and difficulty with taking drugs daily. Another was competing demands, such as out-of-station work-related travel, which caused missed clinic visits [29]. These challenges can be addressed through PrEP adherence counseling and tailored services to address individual needs. However, the impact of PrEP adherence counseling can be negated by risky sexual behavior, alcohol use, younger age, and length of PrEP use [30]. It will be important to explore other ways to improve adherence to PrEP, including understanding the risk-taking profile of clients on PrEP. Past demonstration studies have indicated that adherence to PrEP decreases with time [8, 30].

The low HIV seroconversion rate in this study may be the result of the good adherence among the HIV-infected partners, resulting in viral suppression and reduced transmission risk to the HIV-uninfected partner [31] despite poor adherence to PrEP. Although we are unable to decipher the relative contribution of PrEP and adherence of HIV-positive participants to ART to the low seroconversion rate, the study finding suggests that preventive treatment is an important strategy for lowering the risk of seroconversion in HIV-1 serodiscordant couples in Nigeria. This study suggests that PrEP plays a complementary role in the prevention of HIV-1 seroconversion for HIV serodiscordant couples in our country, as highlighted in a modeling study [32].

## Conclusion

In this study of three delivery models for promoting access of HIV-1 serodiscordant couples to PrEP in Nigeria, the number of clients lost to follow up increased with time. However, client retention was higher in an outpatient model than in the ART clinic and decentralized models. The incidence of HIV seroconversion was slightly higher at the decentralized service delivery model than at the outpatient delivery model and the ART clinic delivery model; this difference may be due to the higher sexual risk behavior of study participants at the decentralized model

rather than the type of service delivery model. The study findings imply that any of the models can effectively deliver PrEP services. Future studies should be directed at understanding how risk behaviour affects adherence to PrEP and implications of the behaviour for risk-reduction counseling for HIV serodiscordant couples.

## Supporting information

**S1 Checklist. CONSORT 2010 checklist of information to include when reporting a randomised trial**[*]**.**
(DOCX)

**S1 File.**
(PDF)

**S2 File.**
(PDF)

**S1 Fig.**
(JPG)

**S2 Fig.**
(JPG)

## Acknowledgments

The contributions of Drs Kwasi Torpey and Titi Badru to the review and editing of this manuscript are acknowledged.

## Author Contributions

**Conceptualization:** Atiene Sagay, Hadiza Kamofu, John Idoko.

**Data curation:** Oliver Ezechi, Matthias Alagi, Chidi Nweneka.

**Formal analysis:** Ayodeji Oginni, Nancin Dadem, Matthias Alagi, John Idoko.

**Funding acquisition:** John Idoko.

**Investigation:** Grace Kolawole, Nkiru Ezeama, Nancin Dadem, Evaristus Afiadigwe, Chidi Nweneka.

**Methodology:** Morenike Oluwatoyin Folayan, Oliver Ezechi, Grace Kolawole, Matthias Alagi, Rose Aguolu, Hadiza Kamofu.

**Project administration:** Sani Aliyu, Nkiru Ezeama, James Anenih, Etim Ekanem, Evaristus Afiadigwe, Rose Aguolu, Tinuade Oyebode, Atiene Sagay, Hadiza Kamofu, John Idoko.

**Resources:** Grace Kolawole, Nancin Dadem, James Anenih, Rose Aguolu, Alero Babalola-Jacobs.

**Supervision:** Morenike Oluwatoyin Folayan, Nkiru Ezeama, Etim Ekanem, Evaristus Afiadigwe, Tinuade Oyebode, Alero Babalola-Jacobs, Atiene Sagay, Chidi Nweneka.

**Validation:** Ayodeji Oginni.

**Writing – original draft:** Morenike Oluwatoyin Folayan, Ayodeji Oginni, Oliver Ezechi.

**Writing – review & editing:** John Idoko.

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
