## [Decision Letter · Decision Letter 0]

23 Sep 2020

PONE-D-20-09560

EFFECTIVENESS OF THREE DELIVERY MODELS FOR PROMOTING ACCESS TO PRE-EXPOSURE PROPHYLAXIS IN HIV-1 SERODISCORDANT COUPLES IN NIGERIA

PLOS ONE

Dear Dr.Idoko

Thank you for submitting your manuscript to PLOS ONE. After careful consideration, we feel that it has merit but does not fully meet PLOS ONE’s publication criteria as it currently stands. Therefore, we invite you to submit a revised version of the manuscript that addresses the points raised during the review process.

We look forward to receiving your revised manuscript.

Kind regards,

Nitika Pant Pai, MD., MPH., PhD

Academic Editor

PLOS ONE

5. Please upload a copy of Figure 2 and 4, to which you refer in your text on page 24. (there are 3 figures captioned as Figure 1.)  If the figure is no longer to be included as part of the submission please remove all reference to it within the text.

6.Thank you for submitting your clinical trial to PLOS ONE and for providing the name of the registry and the registration number. The information in the registry entry suggests that your trial was registered after patient recruitment began. PLOS ONE strongly encourages authors to register all trials before recruiting the first participant in a study.

1) your reasons for your delay in registering this study (after enrolment of participants started);

2) confirmation that all related trials are registered by stating: “The authors confirm that all ongoing and related trials for this drug/intervention are registered”.

Please also ensure you report the date at which the ethics committee approved the study as well as the complete date range for patient recruitment and follow-up in the Methods section of your manuscript.

Editor comments: 

While we all agree that the rationale for the proposed exploration is sound, however, we would like to know more about the design that you chose, the arms that you compared, the role of confounding and exploration of confounding with statistical methods (post hoc or pre hoc) and the conclusions that you drew from them.

The proposed design will be better addressed as a cohort design where you could explore confounding with propensity scores in a regression model, to begin with. A cohort design would allow exploration of multiple hypotheses with multiple confounding factors affecting it. The exploration looks like a cohort- but its written up like a trial.

If you choose a TRIAL use CONSORT, if you did a cohort study, use STROBE, to report the study.

The exploration is pitched as a randomized  trial comparison.Hypotheses, objectives, endpoints and comparisons need to be clearly defined. 3 arms (service delivery models) make it difficult power wise. Besides, for multiple outcomes (need the alpha to be divided by the number of comparisons (3) but that seems to be not the case.

We would like you to address all the points raised by us, so that we can get a fair sense of what you did, and how you proposed to handle comparisons.

if you could respond to our comments below, we would be happy to consider it.

Reviewers' comments:

Reviewer's Responses to Questions

**Comments to the Author**

1. Is the manuscript technically sound, and do the data support the conclusions?

Reviewer #2: Partly

2. Has the statistical analysis been performed appropriately and rigorously? 

Reviewer #2: No

3. Have the authors made all data underlying the findings in their manuscript fully available?

Reviewer #2: Yes

4. Is the manuscript presented in an intelligible fashion and written in standard English?

Reviewer #2: Yes

5. Review Comments to the Author

Reviewer #1: The objective of this study was to evaluate the effectiveness of three models for pre-exposure prophylaxis (PrEP) service delivery to HIV-1 serodiscordant couples in Nigeria.

Below are some comments:

- The experimental unit of this study was site as randomization was done at site level. However there was only a single site for each treatment arm. This essentially makes the study arm un-testable utilizing the randomization (i.e. no possible inference in a randomized setting based on sampling unit). Testing the 3 arms on patient level make this study merely an observational/cohort study and prone to the problems known in such study (see below).

- The current study is in fact a cohort study with 3 different arms/cohorts. As generally known such study is prone to confounding factor problems when attempting to compare between the arms/cohorts. Had the impact of external variables such as age, viral load of partner, region, coital act, use of condom on uptake of PrEP and TasP and on HIV sero-conversion been assessed and incorporate as adjustment factors into the model? This kind of adjustment is needed (for example using Cox regression mode adjusting for those factors instead of simple logrank test), particularly since the analysis suggested the imbalance of these factors across the arms (there were significant differences in the mean age, and the sex, educational level and wealth distribution of the study participants per model). Other factors that need to be considered for the adjustment are the number of unprotected sex acts in the prior month and the number of participants who had sex outside the relationship in the prior month.

- The sample size of the study was calculated based on 2-sided alpha = 5%, when in fact the number of comparisons between arms was 3. It simply suggests that number of comparison was not taking into account at the design level, i.e. no adjustment to the level of alpha. How do you justify this?

- For the survival analysis, the event of interest was the discontinuation of couple’s visit. Did you have any case when only one person of the couple visited? How do you handle such case in term of event or censoring?

- What was the visit window in all the 3 arms? How frequent was the couples seen/assessed in these 3 different arms? This visit information is sensitive when comparing the KM curves.

- Lines 291-292: ‘The HIV-1 positive incidence before initiation of the study was compared with the HIV-1 incidence at the end of the study per model. The pre-study initiation data at the study site was collected 12 months prior to study initiation’. Apparently the pre-study incidence was collected retrospectively. How did you handle potential bias due to this retrospective collection? How did you ensure the validity of the comparison between pre and end study incidence?

- Adherence (lines 348-359). ‘At months 3, 6, 9 and 12 follow-up visits, 152 (74.5%), 117 (64.3%), 77 (47.5%) and 71 (32.1%) subjects continued to receive PrEP, respectively …’, as this was a self-reporting adherence, had there any mechanism in the study established to verify this?

- Discussion, lines 374-375 stated: ‘Results of the study indicate that although the rate of retention of study participants on PrEP did not differ by model adherence to PrEP differed according to model: …’ yet at the conclusion stated: ’However, client retention was significantly higher in an outpatient model than in the ART clinic and decentralized models.’ How do you explain this discrepancy?

Reviewer #2: This manuscript describes a demonstration project of PrEP among HIV serodiscordant couples in Nigeria.

Broadly, the work seems like it was reasonably done. The premise of the project is strong. The analyses are hard to follow, however, and important concerns arise.

Major comments:

Comparisons across models. The design of the study is not particularly clear. It seems to be a cohort, with three different models at three locations. In at least one place, these models are described as randomized (Line 136). However, a randomized comparison of n=3 is not a very powerful design and does not permit a robust comparison. There are many factors – staffing of clinic, catchment of clinic, etc. – that undoubtedly influence outcomes in a study like this. The presentation of results and interpretation should recognize this much more. To have done a true robust comparison would have been randomly assigning probably a dozen or more clinics to the different delivery approaches. Instead, simply acknowledging that three models were used and comparisons across them are subject to bias and confounding that is hard to assess is probably fine for a pilot demonstration like this.

Abstract. Per the point above, the results should focus more on the findings and less on the comparisons across groups. The retention and adherence numbers aren’t even described – just p-values – which is exactly the opposite of what is needed.

Retention. The retention numbers are difficult to follow, in part because different numbers are presented in different ways (Table 2, Figure 1, Figure 2). Figure 2 in particular looks like terrible retention.

HIV incidence. Incident HIV is described as the primary outcome (like 256), which seems impossible – the study could not have been powered, either in each model or in all three combined, to assess HIV incidence. The incidence analyses are difficult to interpret – the text is not clear what is HIV incidence before the study period (and who were these individuals before the study period? Are they beyond the 297 followed?) and what is in the study period. Figure 4 is not helpful, and the analyses associated are terribly underpowered. The authors might just be better off saying there was 1 HIV infection in the 297 couples, in a couple who had done 6 months of ART and stopped PrEP (and whether viral load information was known for the partner living first with HIV).

Minor comments:

Lines 61-62. The statement that PrEP has reduced HIV at the population level is true, but the placement of this sentence seems off – the authors may be instead referring to clinical trial populations rather than truly at the population level. A reference would be good too.

Line 111. Why were pregnant and lactating women excluded?

Line 176. What is the Partners PrEP Study approach?

6. PLOS authors have the option to publish the peer review history of their article (what does this mean?). If published, this will include your full peer review and any attached files.

Reviewer #1: No

Reviewer #2: No

---

## [Author Response · Author response to Decision Letter 0]

12 Jun 2021

Reviewer's Responses to Questions

Comments to the Author

1. Is the manuscript technically sound, and do the data support the conclusions?

Reviewer #2: Partly

Response to comment: The manuscript has been revised to better describe the study design.

2. Has the statistical analysis been performed appropriately and rigorously?

Reviewer #2: No

Response to comment: The statistical analysis has been revised and appropriate test statistics applied to the analysis.

3. Have the authors made all data underlying the findings in their manuscript fully available?

Reviewer #2: Yes

4. Is the manuscript presented in an intelligible fashion and written in standard English?

Reviewer #2: Yes

5. Review Comments to the Author

Reviewer #1: The objective of this study was to evaluate the effectiveness of three models for pre-exposure prophylaxis (PrEP) service delivery to HIV-1 serodiscordant couples in Nigeria.

Below are some comments:

- The experimental unit of this study was site as randomization was done at site level. However there was only a single site for each treatment arm. This essentially makes the study arm un-testable utilizing the randomization (i.e. no possible inference in a randomized setting based on sampling unit). Testing the 3 arms on patient level make this study merely an observational/cohort study and prone to the problems known in such study (see below).

Response: Thank you for your review and comment. The study seeks to identify which of the three major HIV treatment delivery models operational in Nigeria will be suitable to deliver PrEP as an intervention to HIV-1 serodiscordant couples. This informed the choice of study design and selection of sites that have recorded high numbers of serodiscordant couples. In addition, eligible and consenting participants were recruited into each model based on the location they access HIV service.

- The current study is in fact a cohort study with 3 different arms/cohorts. As generally known such study is prone to confounding factor problems when attempting to compare between the arms/cohorts. Had the impact of external variables such as age, viral load of partner, region, coital act, use of condom on uptake of PrEP and TasP and on HIV sero-conversion been assessed and incorporate as adjustment factors into the model? This kind of adjustment is needed (for example using Cox regression mode adjusting for those factors instead of simple logrank test), particularly since the analysis suggested the imbalance of these factors across the arms (there were significant differences in the mean age, and the sex, educational level and wealth distribution of the study participants per model). Other factors that need to be considered for the adjustment are the number of unprotected sex acts in the prior month and the number of participants who had sex outside the relationship in the prior month.

Response: Your comment is appreciated. We recognize some of these limitations and have included Cox regression analysis to examine multivariate factors across the models.

- The sample size of the study was calculated based on 2-sided alpha = 5%, when in fact the number of comparisons between arms was 3. It simply suggests that number of comparison was not taking into account at the design level, i.e. no adjustment to the level of alpha. How do you justify this?

Response: Thank you. The sample size formula for comparing two or more proportions was used in the study to determine the minimum sample size per group at 110. After adjusting for attrition the sample size came to 130 participants per group.

- For the survival analysis, the event of interest was the discontinuation of couple’s visit. Did you have any case when only one person of the couple visited? How do you handle such case in term of event or censoring?

Response: Thank you. Yes we had few cases when only one of the couples visited during the follow up visit period. Per protocol such couples were censored to have discontinued the study. They were adjudged as being ineligible to continue the study. 

- What was the visit window in all the 3 arms? How frequent was the couples seen/assessed in these 3 different arms? This visit information is sensitive when comparing the KM curves.

Response: The visit window were same for the three models. The couples were seen at month 0, 3, 6, 9, and 12.

- Lines 291-292: ‘The HIV-1 positive incidence before initiation of the study was compared with the HIV-1 incidence at the end of the study per model. The pre-study initiation data at the study site was collected 12 months prior to study initiation’. Apparently the pre-study incidence was collected retrospectively. How did you handle potential bias due to this retrospective collection? How did you ensure the validity of the comparison between pre and end study incidence?

Response: Thank you. The facility HIV records were the source of the retrospective data on seroconversion rate among serodiscordant couples for the pre-study period. The comparison was made pre-study and post-study as incidence per 100 person years. 

- Adherence (lines 348-359). ‘At months 3, 6, 9 and 12 follow-up visits, 152 (74.5%), 117 (64.3%), 77 (47.5%) and 71 (32.1%) subjects continued to receive PrEP, respectively …’, as this was a self-reporting adherence, had there any mechanism in the study established to verify this?

Response: Thank you so much for this comment. A number of mechanisms were established in the study to verify self-reported adherence among the participants. Firstly, among the HIV-1 infected partners that are on ART viral load test were conducted to ascertain if they are achieving viral suppression; as expected if they are adherent to their ART. In addition, HIV-1 infected partners ARV drug dispensing records were also monitored for missed appointments, default and lost to follow up. On the other hand, for HIV-1 uninfected partner self reported adherence was verified through use of MEMSCap, a drug dispensing bottle embedded with a microchip to detect the number of times and days participant open the bottle to take PrEP medication. Also, biomarker test for presence of ARvmetabolites was conducted for participants on PrEP.

- Discussion, lines 374-375 stated: ‘Results of the study indicate that although the rate of retention of study participants on PrEP did not differ by model adherence to PrEP differed according to model: …’ yet at the conclusion stated: ’However, client retention was significantly higher in an outpatient model than in the ART clinic and decentralized models.’ How do you explain this discrepancy?

Response: Thank you so much, the correction has been effected.

Reviewer #2: This manuscript describes a demonstration project of PrEP among HIV serodiscordant couples in Nigeria.

Broadly, the work seems like it was reasonably done. The premise of the project is strong. The analyses are hard to follow, however, and important concerns arise.

Major comments:

Comparisons across models. The design of the study is not particularly clear. It seems to be a cohort, with three different models at three locations. In at least one place, these models are described as randomized (Line 136). However, a randomized comparison of n=3 is not a very powerful design and does not permit a robust comparison. There are many factors – staffing of clinic, catchment of clinic, etc. – that undoubtedly influence outcomes in a study like this. The presentation of results and interpretation should recognize this much more. To have done a true robust comparison would have been randomly assigning probably a dozen or more clinics to the different delivery approaches. Instead, simply acknowledging that three models were used and comparisons across them are subject to bias and confounding that is hard to assess is probably fine for a pilot demonstration like this.

Response: Thank you for your review and comment. The study seeks to identify which of the three major HIV treatment delivery models operational in Nigeria will be suitable to deliver PrEP as an intervention to HIV-1 serodiscordant couples. This informed the choice of study design and selection of sites that have recorded high numbers of serodiscordant couples. The comparison is made across the three models, such that each model served as the control for the other two.

Abstract. Per the point above, the results should focus more on the findings and less on the comparisons across groups. The retention and adherence numbers aren’t even described – just p-values – which is exactly the opposite of what is needed.

Response: Thank you, the correction has been effected.

Retention. The retention numbers are difficult to follow, in part because different numbers are presented in different ways (Table 2, Figure 1, Figure 2). Figure 2 in particular looks like terrible retention.

Response: Thank you. We admit that retention in the study declined with time and it is difficult to have same numbers for different time period. The result has been represented in a simpler way in the revision.

HIV incidence. Incident HIV is described as the primary outcome (like 256), which seems impossible – the study could not have been powered, either in each model or in all three combined, to assess HIV incidence. The incidence analyses are difficult to interpret – the text is not clear what is HIV incidence before the study period (and who were these individuals before the study period? Are they beyond the 297 followed?) and what is in the study period. Figure 4 is not helpful, and the analyses associated are terribly underpowered. The authors might just be better off saying there was 1 HIV infection in the 297 couples, in a couple who had done 6 months of ART and stopped PrEP (and whether viral load information was known for the partner living first with HIV).

Response: Thank you for your review comment. The result have been represented for clarity.

Minor comments:

Lines 61-62. The statement that PrEP has reduced HIV at the population level is true, but the placement of this sentence seems off – the authors may be instead referring to clinical trial populations rather than truly at the population level. A reference would be good too.

Response: Thank you for pointing this out, references has been cited.

Line 111. Why were pregnant and lactating women excluded?

Response: Thank you. Pregnant and lactating women were excluded because the teratogenic effect and effect on the breastfeeding child of the PrEP medication tenofovir was not well known at the time of the study.

Line 176. What is the Partners PrEP Study approach?

Response: Thank you. The Partners PrEP Study is a double-blind, placebo-controlled, phase II clinical trial to assess the safety and efficacy of oral PrEP for the prevention of HIV infection, using antiretroviral medication tenofovir (TDF), either alone or in combination with emtricitabine (FTC/TDF). The trial demonstrated the benefit of tenofovir and recommended the discontinuation of placebo. The Partners PrEP Study approach is a group of demonstration study on how to deliver PrEP effectively for populations at greatest risk for HIV. The study is funded by Bill & Melinda Gates Foundation. The drugs are donated by Gilead Sciences Incorporated.

---

## [Decision Letter · Decision Letter 1]

10 Nov 2021

PONE-D-20-09560R1EFFECTIVENESS OF THREE DELIVERY MODELS FOR PROMOTING ACCESS TO PRE-EXPOSURE PROPHYLAXIS IN HIV-1 SERODISCORDANT COUPLES IN NIGERIAPLOS ONE

Dear Dr. Idoko,

Thank you for submitting your manuscript to PLOS ONE. After careful consideration, we feel that it has merit but does not fully meet PLOS ONE’s publication criteria as it currently stands. Therefore, we invite you to submit a revised version of the manuscript that addresses the points raised during the review process. In particular, there are some very significant issues with regards to sample size, randomization, and methods.  These issues must be addressed to accept this paper for publication.  Please submit your revised manuscript by Dec 25 2021 11:59PM. If you will need more time than this to complete your revisions, please reply to this message or contact the journal office at plosone@plos.org. Please include the following items when submitting your revised manuscript:A rebuttal letter that responds to each point raised by the academic editor and reviewer(s). You should upload this letter as a separate file labeled 'Response to Reviewers'.A marked-up copy of your manuscript that highlights changes made to the original version. You should upload this as a separate file labeled 'Revised Manuscript with Track Changes'.An unmarked version of your revised paper without tracked changes. You should upload this as a separate file labeled 'Manuscript'.

We look forward to receiving your revised manuscript.

Kind regards,

Matt A Price

Academic Editor

PLOS ONE

Reviewers' comments:

Reviewer's Responses to Questions

**Comments to the Author**

1. If the authors have adequately addressed your comments raised in a previous round of review and you feel that this manuscript is now acceptable for publication, you may indicate that here to bypass the “Comments to the Author” section, enter your conflict of interest statement in the “Confidential to Editor” section, and submit your "Accept" recommendation.

Reviewer #1: (No Response)

Reviewer #2: All comments have been addressed

2. Is the manuscript technically sound, and do the data support the conclusions?

Reviewer #1: Partly

Reviewer #2: Yes

3. Has the statistical analysis been performed appropriately and rigorously? 

Reviewer #1: No

Reviewer #2: Yes

4. Have the authors made all data underlying the findings in their manuscript fully available?

Reviewer #1: Yes

Reviewer #2: Yes

5. Is the manuscript presented in an intelligible fashion and written in standard English?

Reviewer #1: Yes

Reviewer #2: Yes

6. Review Comments to the Author

Reviewer #1: Some of the major not properly addressed, namely:

- The experimental unit of this study was site as randomization was done at site level. However there was only a single site for each treatment arm. This essentially makes the study arm un-testable utilizing the randomization (i.e. no possible inference in a randomized setting based on sampling unit). Testing the 3 arms on patient level make this study merely an observational/cohort study and prone to the problems known in such study (see below).

Response: Thank you for your review and comment. The study seeks to identify which of the three major HIV treatment delivery models operational in Nigeria will be suitable to deliver PrEP as an intervention to HIV-1 serodiscordant couples. This informed the choice of study design and selection of sites that have recorded high numbers of serodiscordant couples. In addition, eligible and consenting participants were recruited into each model based on the location they access HIV service.

Re: The above response does not address the comments appropriately. In the current study, your randomization can not be used as a basis of inference. Please address this question.

- Original comment: The current study is in fact a cohort study with 3 different arms/cohorts. As generally known such study is prone to confounding factor problems when attempting to compare between the arms/cohorts. Had the impact of external variables such as age, viral load of partner, region, coital act, use of condom on uptake of PrEP and TasP and on HIV sero-conversion been assessed and incorporate as adjustment factors into the model? This kind of adjustment is needed (for example using Cox regression mode adjusting for those factors instead of simple logrank test), particularly since the analysis suggested the imbalance of these factors across the arms (there were significant differences in the mean age, and the sex, educational level and wealth distribution of the study participants per model). Other factors that need to be considered for the adjustment are the number of unprotected sex acts in the prior month and the number of participants who had sex outside the relationship in the prior month.

Response: Your comment is appreciated. We recognize some of these limitations and have included Cox regression analysis to examine multivariate factors across the models.

Re: This comment is not addressed properly. The Cox model which is presented in the current article does not do any comparison between three major HIV treatment delivery models.

- Original comment: The sample size of the study was calculated based on 2-sided alpha = 5%, when in fact the number of comparisons between arms was 3. It simply suggests that number of comparison was not taking into account at the design level, i.e. no adjustment to the level of alpha. How do you justify this?

Response: Thank you. The sample size formula for comparing two or more proportions was used in the study to determine the minimum sample size per group at 110. After adjusting for attrition the sample size came to 130 participants per group.

Re: This comment is not addressed properly. Justification of not penalizing alpha despite multiple comparisons is needed.

- Original comment(Lines 291-292): ‘The HIV-1 positive incidence before initiation of the study was compared with the HIV-1 incidence at the end of the study per model. The pre-study initiation data at the study site was collected 12 months prior to study initiation’.

Apparently the pre-study incidence was collected retrospectively. How did you handle potential bias due to this retrospective collection? How did you ensure the validity of the comparison between pre and end study incidence?

Response: Thank you. The facility HIV records were the source of the retrospective data on seroconversion rate among serodiscordant couples for the pre-study period. The comparison was made pre-study and post-study as incidence per 100 person years.

Re: This comment is not addressed properly. Data collected retrospectively and data collected prospectively are 2 very different data in term of quality. This should be made clear in the article including its consequence.

Reviewer #2: My comments have been addressed

7. PLOS authors have the option to publish the peer review history of their article (what does this mean?). If published, this will include your full peer review and any attached files.

Reviewer #1: No

Reviewer #2: No

---

## [Author Response · Author response to Decision Letter 1]

8 Feb 2022

Reviewer #1: Some of the major not properly addressed, namely:

- The experimental unit of this study was site as randomization was done at site level. However there was only a single site for each treatment arm. This essentially makes the study arm un-testable utilizing the randomization (i.e. no possible inference in a randomized setting based on sampling unit). Testing the 3 arms on patient level make this study merely an observational/cohort study and prone to the problems known in such study (see below).

Response: Thank you for your review and comment. The study seeks to identify which of the three major HIV treatment delivery models operational in Nigeria will be suitable to deliver PrEP as an intervention to HIV-1 serodiscordant couples. This informed the choice of study design and selection of sites that have recorded high numbers of serodiscordant couples. In addition, eligible and consenting participants were recruited into each model based on the location they access HIV service.

Re: The above response does not address the comments appropriately. In the current study, your randomization can not be used as a basis of inference. Please address this question.

Response: Thank you for your review comment, the study groups have been amended as cohorts.

- Original comment: The current study is in fact a cohort study with 3 different arms/cohorts. As generally known such study is prone to confounding factor problems when attempting to compare between the arms/cohorts. Had the impact of external variables such as age, viral load of partner, region, coital act, use of condom on uptake of PrEP and TasP and on HIV sero-conversion been assessed and incorporate as adjustment factors into the model? This kind of adjustment is needed (for example using Cox regression mode adjusting for those factors instead of simple logrank test), particularly since the analysis suggested the imbalance of these factors across the arms (there were significant differences in the mean age, and the sex, educational level and wealth distribution of the study participants per model). Other factors that need to be considered for the adjustment are the number of unprotected sex acts in the prior month and the number of participants who had sex outside the relationship in the prior month.

Response: Your comment is appreciated. We recognize some of these limitations and have included Cox regression analysis to examine multivariate factors across the models.

Re: This comment is not addressed properly. The Cox model which is presented in the current article does not do any comparison between three major HIV treatment delivery models.

Response: Thank you. A Cox survival analysis with adjusted Hazard ratio has been conducted to compare among the three treatment groups, please refer table 4.

- Original comment: The sample size of the study was calculated based on 2-sided alpha = 5%, when in fact the number of comparisons between arms was 3. It simply suggests that number of comparison was not taking into account at the design level, i.e. no adjustment to the level of alpha. How do you justify this?

Response: Thank you. The sample size formula for comparing two or more proportions was used in the study to determine the minimum sample size per group at 110. After adjusting for attrition the sample size came to 130 participants per group.

Re: This comment is not addressed properly. Justification of not penalizing alpha despite multiple comparisons is needed.

Response: Thank you. A Bonferroni correction factor of 3 has been applied to the alpha to account for the three treatment groups. 

- Original comment(Lines 291-292): ‘The HIV-1 positive incidence before initiation of the study was compared with the HIV-1 incidence at the end of the study per model. The pre-study initiation data at the study site was collected 12 months prior to study initiation’.

Apparently the pre-study incidence was collected retrospectively. How did you handle potential bias due to this retrospective collection? How did you ensure the validity of the comparison between pre and end study incidence?

Response: Thank you. The facility HIV records were the source of the retrospective data on seroconversion rate among serodiscordant couples for the pre-study period. The comparison was made pre-study and post-study as incidence per 100 person years.

Re: This comment is not addressed properly. Data collected retrospectively and data collected prospectively are 2 very different data in term of quality. This should be made clear in the article including its consequence.

Response: Thank you. Data collected retrospectively before the study are prone to data quality issues such as completeness, accuracy and precision of operational definitions.

---

## [Decision Letter · Decision Letter 2]

9 Mar 2022

PONE-D-20-09560R2EFFECTIVENESS OF THREE DELIVERY MODELS FOR PROMOTING ACCESS TO PRE-EXPOSURE PROPHYLAXIS IN HIV-1 SERODISCORDANT COUPLES IN NIGERIAPLOS ONE

Dear Dr. Idoko,

Thank you for submitting your manuscript to PLOS ONE. After careful consideration, we feel that it has merit but does not fully meet PLOS ONE’s publication criteria as it currently stands. Therefore, we invite you to submit a revised version of the manuscript that addresses the points raised during the review process.

We look forward to receiving your revised manuscript.

Kind regards,

Matt A Price

Academic Editor

PLOS ONE

Journal Requirements:

Reviewers' comments:

Reviewer's Responses to Questions

**Comments to the Author**

1. If the authors have adequately addressed your comments raised in a previous round of review and you feel that this manuscript is now acceptable for publication, you may indicate that here to bypass the “Comments to the Author” section, enter your conflict of interest statement in the “Confidential to Editor” section, and submit your "Accept" recommendation.

Reviewer #1: All comments have been addressed

2. Is the manuscript technically sound, and do the data support the conclusions?

Reviewer #1: Yes

3. Has the statistical analysis been performed appropriately and rigorously? 

Reviewer #1: Yes

4. Have the authors made all data underlying the findings in their manuscript fully available?

Reviewer #1: Yes

5. Is the manuscript presented in an intelligible fashion and written in standard English?

Reviewer #1: (No Response)

6. Review Comments to the Author

Reviewer #1: Thanks for addressing the comments. Here is a minor one on time to event definition:

Lines 250-253:

The event of interest was the discontinuation of couple’s visit. Those whose duration of stay in the study was < 12 months were categorized as discontinued (coded as 1), while those whose period of stay in the study was ≥12months were categorized as those who remained (coded as 0) in the study ....

I presume this define how the survival response variable is coded. The current description is more like a binary variable when in fact it is a time to event. I suggest to use the following:

The event of interest was the discontinuation of couple’s visit. Those who discontinued before 12 months were recorded to have an event at the time of discontinuation. Those whose period of stay longer than 12 months were censored at 12 months.

7. PLOS authors have the option to publish the peer review history of their article (what does this mean?). If published, this will include your full peer review and any attached files.

Reviewer #1: No

---

## [Author Response · Author response to Decision Letter 2]

19 Mar 2022

Response to the Reviewer comment

Thank you very kindly for the comment, your suggestion has been incorporated into the revised manuscript.

---

## [Editor Report · Decision Letter 3]

21 Apr 2022

EFFECTIVENESS OF THREE DELIVERY MODELS FOR PROMOTING ACCESS TO PRE-EXPOSURE PROPHYLAXIS IN HIV-1 SERODISCORDANT COUPLES IN NIGERIA

PONE-D-20-09560R3

Dear Dr. Idoko,

We’re pleased to inform you that your manuscript has been judged scientifically suitable for publication and will be formally accepted for publication once it meets all outstanding technical requirements.

Kind regards,

Matt A Price

Academic Editor

PLOS ONE
---

## [Editor Report · Acceptance letter]

26 Apr 2022

PONE-D-20-09560R3 

Effectiveness of three delivery models for promoting access to pre-exposure prophylaxis in HIV-1 serodiscordant couples in Nigeria 

Dear Dr. Idoko:

I'm pleased to inform you that your manuscript has been deemed suitable for publication in PLOS ONE. Congratulations! Your manuscript is now with our production department. 

Kind regards, 

on behalf of

Dr. Matt A Price 

Academic Editor

PLOS ONE